# VIRO: Efficient and Robust Neuro-Symbolic Reasoning with Verification for Referring Expression Comprehension

## Abstract

Referring Expression Comprehension (REC) aims to localize the image region corresponding to a natural language query. To handle complex queries, recent work has focused on compositional reasoning, with advances in Large Language Models (LLMs) and Vision Language Models (VLMs) enabling the decomposition of queries into executable programs within reasoning pipelines. However, existing approaches implicitly assume the target is always present, forcing the model to output a result even when no valid referent exists. Moreover, multi-step reasoning processes often result in high computational costs, limiting their application in real-time scenarios. To address this limitation, we propose Verification-Integrated Reasoning Operators (VIRO), which integrate operator-level verification into a neuro-symbolic pipeline, enabling abstention and the explicit handling of no-target cases. Each operator performs a reasoning step and verifies its own execution, including a lightweight CLIP-based filter with minimal computational overhead, and logical verification for spatial and relational constraints. Experimental results demonstrate that our framework achieves strong robustness in no-target cases, achieving 61.1% balanced accuracy, while showing state-of-the-art accuracy on standard REC benchmarks, compared to compositional baselines. Our neuro-symbolic pipeline also shows superior computational efficiency, high reliability with a program failure rate of just 0.3%, and scalability—achieved by decoupling program generation from execution.

## 1 Introduction

Humans often rely on language to navigate and interpret visual environments, using referring expressions, such as "the blue cup on the wooden table," to pinpoint specific objects in complex scenes. This fundamental human skill is formalized in the vision-language task of Referring Expression Comprehension (REC), where the goal is to localize a target object in an image based on a natural language description (Qiao et al., 2020). This task has broad applicability in artificial intelligence (AI), including vision-language navigation (Wang et al., 2021; Zhang et al., 2024) and human-robot interaction (Shridhar et al., 2022; Jin et al., 2025), and text-to-image retrieval (Lee et al., 2024).

Early REC approaches have relied on supervised end-to-end learning that directly map textual queries to the corresponding regions (Yu et al., 2018; Kamath et al., 2021; Yan et al., 2023). With the advent of large language models (LLMs), the field has shifted toward compositional strategies that parse a natural language description into structured semantic components (Subramanian et al., 2022; Han et al., 2024; Shen et al., 2024; Chen & Chen, 2025). By decomposing queries into smaller semantic units, this approach enables systematic reasoning over complex relationships between objects, attributes, and spatial relations (Surís et al., 2023; Ke et al., 2024; Cai et al., 2025), and offers flexibility in interpreting diverse linguistic patterns. Moreover, recent advancements in Open-Vocabulary Detectors (OVDs) (Li et al., 2022; Liu et al., 2024; Xiao et al., 2024), with their strong zero-shot capabilities, enable a more direct mapping of language to visual evidence.

However, these approaches assume that a valid referent is present in every image, forcing a prediction even in no-target scenes. In parallel, compositional methods (Surís et al., 2023; Ke et al., 2024; Cai et al., 2025; Chen & Chen, 2025) rely on open-vocabulary detectors; while effective on

Figure 1: An illustrative comparison of between previous REC methods and our VIRO framework in no-target cases. Previous REC methods (left) are forced to output a prediction, even when the query cannot be grounded in the image, due to the lack of a mechanism for eliminating incorrect candidates. In contrast, our VIRO framework (right) rejects invalid cases: (i) FIND identifies that there is no elephant in the image (top); (ii) FIND_DIRECTION identifies the person is not positioned to the left of the elephant (bottom). VIRO can terminate early if there are no candidates.

unseen categories, these detectors often hallucinate high-confidence false positives for non-existent objects, which exacerbate the problem and furthermore, propagate through relational/spatial modules. While recent works by He et al. (2023) and Liu et al. (2023) have acknowledged the existence of no-target cases in their datasets, their solutions rely on task-specific supervision. This dependency on supervised training limits the models' applicability in more generalized scenarios where such specific training data is unavailable. Consequently, algorithmic progress in zero-shot REC with explicit handling of no-target cases remains a largely unexplored area of research.

To tackle the challenges, we introduce the Verification-Integrated Reasoning Operators (VIRO), a neuro-symbolic framework designed to explicitly handle such scenarios through operator-level verification. Our pipeline builds on neuro-symbolic reasoning, decomposing natural language descriptions using LLM into a sequence of executable operators, as illustrated in Figure 1, and executes them sequentially. Crucially, VIRO embeds verification within each reasoning step, allowing operators to abstain from forced predictions and to terminate early when conditions are not met. This verification process includes two key components: a lightweight CLIP-based filter with minimal computational overhead that suppresses high-confidence false positives from open-vocabulary detectors, and logical checks that strictly enforce spatial and relational constraints, yielding robustness in no-target scenarios.

Our framework is evaluated on both no-target and standard REC scenarios. On the gRefCOCO no-target dataset (He et al., 2023), it exhibits strong robustness, achieving state-of-the-art accuracy when compared to compositional baselines across standard REC benchmarks, including RefCOCO/+/g. Furthermore, our neuro-symbolic approach achieves high throughput (FPS), a low program failure rate, and enhanced scalability for processing multiple images from a single query.

Our key contributions are summarized as follows:

- We introduce a neuro-symbolic framework built on VIRO, primitive operators that embed verification directly within each reasoning step, enabling explicit handling of no-target scenarios through early termination when verification conditions are not met.

- We design lightweight verification mechanisms including a CLIP-based uncertainty filter to reduce false positives from open-vocabulary detectors and logical verification for spatial reasoning, achieving robust performance without task-specific supervision.

- Our experiments show that superior performance across multiple metrics, including 61.1% balanced accuracy on no-target cases, state-of-the-art results on standard REC benchmarks with a low 0.3% failure rate, exceptional computational efficiency, and scalability through the decoupling of program generation from execution.

## 2 RELATED WORK

**Referring Expression Comprehension.** Early end-to-end REC methods (Yan et al., 2023; Kamath et al., 2021) perform well on seen domains but struggle under shift, as they cannot exploit open-set detectors with strong recognition ability (Liu et al., 2024; Li et al., 2022). To improve zero-shot generalization, the field has shifted toward compositional strategies that first parse textual descriptions into structured semantic units and then align them with candidate proposals for grounding. Subramanian et al. (2022) combines CLIP-based matching with rule-based spatial reasoning, Han et al. (2024) aligns textual and visual triplets through structural similarity, (Shen et al., 2024) fuses heatmaps derived from Vision-Language Pre-training (VLP) models with proposals from an OVD, and Chen & Chen (2025) formalizes queries into structured representations for probabilistic matching. While these approaches enable strong zero-shot transfer, they remain inflexible when queries deviate from pre-defined forms, limiting their ability to handle diverse linguistic inputs.

**Compositional Reasoning REC.** Recent neuro-symbolic approaches for visual grounding offer more flexible compositions than fixed structures. VisProg (Gupta & Kembhavi, 2023) generates abstract pseudo-code that composes vision–language and symbolic modules (e.g., attribute filtering, spatial reasoning), while ViperGPT (Surís et al., 2023) produces less structured code. HYDRA (Ke et al., 2024) introduces iterative reasoning by coupling a planner and reasoner with an RL agent. NAVER (Cai et al., 2025) performs self-correcting inference by circulating information across multiple states. However, these methods assume that the target always exists, forcing a prediction even when no valid object is present. Furthermore, they often require regenerating a program for each new image, which increases computational cost when applying a single query across multiple images. In contrast, our framework addresses these gaps by (i) integrating lightweight verification for robust no-target rejection and (ii) decoupling program generation from execution for greater efficiency.

## 3 METHOD

In Section 3.1, we formally define the REC problem, extended to handle no-target cases. Subsequently, we present the neuro-symbolic reasoning pipeline integrated with VIRO in Section 3.2.

### 3.1 PROBLEM FORMALIZATION

REC aims to localize a region within an image $I$ that corresponds to a given natural language query $Q$. A conventional REC assumes that the target object described by the query is always present in the image. This assumption does not hold in practical applications, such as a visual searching system or a robot searching an object in a building (Zhou et al., 2023; Yokoyama et al., 2024; Zhang et al., 2024; Yin et al., 2025; Gong et al., 2025), where the target is frequently absent from most images. We formalize the output of model as:

$$Y = \begin{cases} B, & \text{if a target exists in } I, \\ \varnothing, & \text{otherwise}. \end{cases} \tag{1}$$

Here, $B = (x, y, w, h) \in \mathbb{R}^4$ denotes an bounding box in pixel coordinates, where $(x, y)$ represents the center coordinates of the box, and $w, h$ are its width and height. The $\varnothing$ denotes the absence of a target, *i.e.*, no object corresponding to the query $Q$ is present in the image.

Table 1: Overview of VIRO. All operators are designed to return a set of verified bounding boxes or an empty set ($\varnothing$) if its condition is not satisfied. Additional operators are provided in Appendix A.1.

| Operator | Input Arguments | Verification Module | Built-in Models |
|---|---|---|---|
| FIND | object_name | CLIP-based verifier | OVD, CLIP |
| PROPERTY | object, attribute | CLIP score | CLIP |
| RELATIVE_DEPTH | object, reference_object, criteria | Relative depth relation | DepthAnything |

### 3.2 A NEURO-SYMBOLIC REASONING PIPELINE

To address the task defined above, our framework employs a two-stage neuro-symbolic pipeline. As illustrated in Figure 2b, the pipeline consists of two main stages: (i) a pre-execution stage that translates the query $Q$ into a symbolic program $P$ (detailed in Section 3.2.2), and (ii) a program execution stage that runs this program $P$ on the image $I$ to localize the referent (detailed in Section 3.2.3). The program $P$ is constructed using verification operators, with additional details on their functionality presented in the following section.

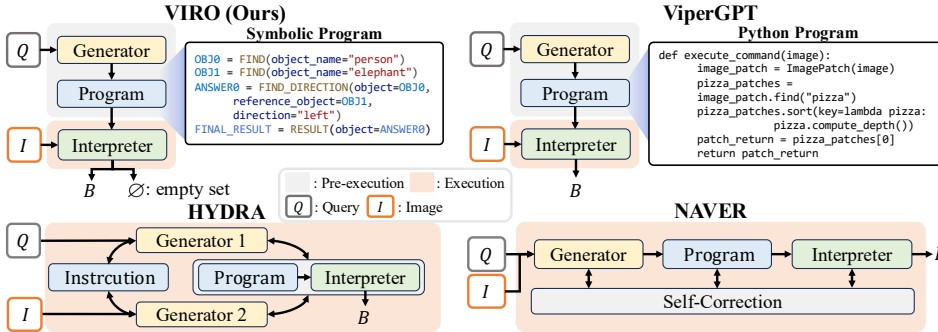

(a) A CLIP-based uncertainty filter within FIND operator to eliminate FPs from OVD proposals.

(b) A pipeline comparison of pre-execution and execution stages.

Figure 2: Overview of VIRO. (a) Details of the FIND operator with a CLIP-based uncertainty filter. (b) The decoupled VIRO pipeline and comparison with compositional reasoning approaches.

### 3.2.1 VERIFICATION-INTEGRATED REASONING OPERATORS (VIRO)

We introduce a finite set of primitive operators, Verification-Integrated Reasoning Operators (VIRO), denoted $\mathcal{O}$ and summarized in Table 1. These operators serve as the foundational building blocks for our neuro-symbolic reasoning pipeline, *i.e.*, $P = (o_1, o_2, \ldots, o_T)$ where $o_t \in \mathcal{O}$ and $T$ denotes the number of program lines. Each operator in $\mathcal{O}$ is designed not only to perform a reasoning step but also to self-verify execution. If an operator determines that its verification condition is not satisfied, the operator returns an empty set ($\varnothing$), enabling early termination of the entire pipeline.

We categorize these operators into four functional categories: (i) **Identification operators**, such as FIND and PROPERTY, which detect candidates and refine entities based on specified attributes; (ii) **Absolute spatial operators**, such as LOCATE, SIZE, ORDER, and ABSOLUTE_DEPTH, which reason about position and scale in absolute terms; (iii) **Relative spatial operators**, such as FIND_DIRECTION, FIND_NEAR, FIND_INSIDE, and RELATIVE_DEPTH, which capture spatial relationships between multiple entities; (iv) **Termination operator**, namely RESULT, which concludes the program by mapping the selected object into the answer space. Further details of each predefined operator, including their arguments, are provided in Table 1. To illustrate how the verification module works, we detail two key examples below.

**Uncertainty Filter in FIND Operator.** The FIND operator takes an argument object_name, a noun phrase $l$ (e.g., "guy") in the query $Q$ (e.g., "A middle guy in red"). It invokes an OVD model $D$ (e.g., Grounding DINO (Liu et al., 2024)) on image $I$ with label $l$ to generate a (possibly empty) set of proposals, *i.e.*, $\{B_j\}_{j=1}^{M} \leftarrow D(I, l)$. While state-of-the-art OVDs offer powerful zero-shot grounding, they can yield high-confidence false positives (FPs) that are visually or semantically similar to $l$, which can propagate error through the reasoning pipeline, as illutrated in Figure 2a.

To mitigate FP proposals, we integrate a lightweight CLIP-based verification module within the FIND operator. This module provides a secondary check on the OVD's proposals by leveraging CLIP's discriminative power of binary classification tasks, effectively refines uncertain outputs' filter while adding minimal computational overhead. Specifically, for each proposal $B_j$, we crop its corresponding image region, denoted as $I_j$. We predefine a bank of $K$ common categories $\mathcal{C} = \{c_1, c_2, \ldots, c_K\}$ that are well-represented in CLIP. The verification score for $B_j$ is the average probability of being the target $l$ when compared one-on-one against each $c_k \in \mathcal{C}$:

$$S(l \mid I_j) = \frac{1}{K} \sum_{k=1}^{K} \frac{\exp(\text{sim}(I_j, l)/\tau)}{\exp(\text{sim}(I_j, l)/\tau) + \exp(\text{sim}(I_j, c_k)/\tau)}, \tag{2}$$

where $\text{sim}(\cdot, \cdot)$ denotes cosine similarity between CLIP image/text embeddings and $\tau > 0$ is a temperature. We accept $B_j$ as $l$ only if $S(l \mid I_j) \geq \delta_l$, where $\delta_l$ is a fixed or label-specific adaptive threshold. There is a trade-off in setting the threshold: when it is set close to $0.5$, true positives (TP) may be filtered out, while FPs may remain unfiltered. Since CLIP can exhibit inherent bias toward labels that were well-represented in its training data, this can affect the accuracy of the thresholding process. To mitigate this, we use ImageNet (Deng et al., 2009) as an auxiliary dataset for per-label threshold calibration. Details of the calibration process are provided in Appendix A.2. To summarize, $S(l|I_j)$ serves as an uncertainty filter, assessing the degree of alignment between the given label $l$ and the regions $I_j$ proposed by OVD, and filtering out uncertain proposals.

**Logical Verification in `FIND_DIRECTION` Operator.** The `FIND_DIRECTION` operator takes three arguments: `object`, `reference_object`, and `direction`. `object` refers to the target object that we are trying to find, while the `reference_object` is the object used as a reference for comparison. It performs a geometric test over all input candidates to verify whether each `object` proposal satisfies the specified spatial relation with respect to at least one `reference_object`.

### 3.2.2 Pre-execution Stage: Symbolic Program Generation

The pre-execution stage translates a given natural-language query $Q$ into a machine-executable symbolic program $P$, which is composed of our primitive operators defined in Section 3.2.1. This translation into a structured format is crucial for ensuring robust execution, as the inherent complexity of natural language make direct machine interpretation unreliable. This process is accomplished through two key components: a program generator that produces an initial program, and a program validator that subsequently ensures its syntactic correctness.

**Program Generation.** We leverage a Large Language Model (LLM) to translate the natural language query $Q$ into a symbolic program $P$. We guide this process using a few-shot prompting strategy as in Gupta & Kembhavi (2023), where the prompt contains a set of exemplars $m$ demonstrating the desired query-to-program mapping:

$$P = \text{LLM}(Q|m) = (o_1, o_2, \ldots, o_T), \quad \text{where } o_t \in \mathcal{O} . \tag{3}$$

**Program Validation.** While powerful, LLMs can occasionally produce programs with syntactic errors such as malformed syntax, wrong operator names, or mismatched arguments. Unlike ViperGPT (Surís et al., 2023) that rely on compiling Python code as shown in Fig 2b, our validator simply checks for conformance to our predefined operator-based grammar. If validation fails, a concise diagnostic is fed back to the LLM in a new prompt to trigger a revision. See Appendix A.3 for details. Furthermore, Symbolic approaches achieve low failure rates, as shown in Section 4.2.2.

### 3.2.3 Execution Stage: Program Interpretation

The program interpreter executes the symbolic program $P = (o_1, o_2, \ldots, o_T)$ sequentially, invoking the corresponding operator at each step. Each operator relies on a built-in model to function properly, and we leverage the following models: GroundingDINO (Liu et al., 2024)/GLIP (Li et al., 2022), CLIP (Radford et al., 2021), and DepthAnything (Yang et al., 2024). Execution continues until either (i) all operators are applied and `RESULT` maps the final candidates to an answer box, or (ii) some operator returns $\varnothing$, yielding a no-target outcome for the current image and immediately terminate. The latter corresponds to an early-exit, which occurs when built-in verification rejects all proposals (e.g., `FIND` identifies there is no valid object; `FIND_DIRECTION` finds no relation as illustrated in Figure 1). This mechanism enables explicit no-target handling and reduce unnecessary computation, as demonstrated in Section 4.3, thereby highlighting the robustness of our verification-integrated design.

### 3.2.4 Decoupled Neuro-Symbolic Approach

As shown in Figure 2b, VIRO adopts a decoupled design that separates program generation from execution. In contrast, methods such as HYDRA (Ke et al., 2024) and NAVER (Cai et al., 2025) entangle program synthesis for a query $Q$ with image execution. Consequently, even when the query is identical across $N$ images, these systems regenerate a reasoning program for each image $I_i$, incurring $N$ separate synthesis operations. VIRO generates the program once and reuses it across all images, enabling low-latency operation. Empirical results are reported in Section 4.2.2.

# 4 EXPERIMENT

In this section, we present a comprehensive evaluation of our VIRO pipeline. The details of the experimental setup are described in Section 4.1. We then present our main results in terms of robustness, efficiency, and scalability in Section 4.2. Following this, we conduct extensive ablation studies in Section 4.3, analyzing the contribution of each component. Finally, Section 4.4 provides an in-depth analysis of our pipeline's behavior through qualitative examples.

## 4.1 EXPERIMENTAL SETUP

**Dataset and Evaluation Metrics.**  We evaluate our framework on both no-target scenarios and standard benchmarks. For no-target setting, we use the gRefCOCO dataset (He et al., 2023), focusing on the no-target split, which contains referring expressions that do not correspond to any object in the image. This setting allows us to directly evaluate the model's ability to suppress incorrect predictions in the absence of a valid target. For standard REC, we evaluate our method on widely used REC benchmarks, including RefCOCO/+ (Yu et al., 2016), and RefCOCOg (Mao et al., 2016).

We evaluate both no-target robustness and standard REC accuracy via **Balanced Accuracy**, defined as $(\text{TPR}+\text{TNR})/2$. It provides a holistic measure by equally weighing the ability to localize existing targets and reject non-existing ones. The component metrics are defined as follows:

- **True Positive Rate (TPR)** = $\frac{TP}{TP+FN}$, measuring accuracy on target-present samples. This is equivalent to the standard Acc@0.5, used in standard REC, where a prediction is a correct if IoU between the predicted and ground-truth bounding boxes exceeds $0.5$.
- **True Negative Rate (TNR)** = $\frac{TN}{TN+FP}$, measuring accuracy on target-absent samples (TP+FP), often referred to as no-target accuracy (N-acc).
- **False Positive Rate (FPR)** = $\frac{FP}{TN+FP} = 1 - \text{TNR}$, quantifying how often the model incorrectly predicts a target in images where none exist.

**Baselines.**  We group existing REC approaches into four categories, according to their underlying assumptions. **Fully supervised REC** includes methods trained end-to-end with full annotations on RefCOCO/+/g, as well as on gRefCOCO no-target dataset for handling no-target cases. **Proposal-based REC** methods first parse the referring expression to extract key linguistic components and then align them with candidate region proposals. Such approaches inherently force the model to select one of the proposals, which makes handling no-target cases intrinsically difficult. **Detector-based REC** leverages large-scale pretrained grounding detectors to directly match textual phrases with image regions in an end-to-end manner, without explicit proposal ranking. For this category, we select GroundingDINO-T (Liu et al., 2024) and GLIP-L (Li et al., 2022) as representative methods. A key criterion for their selection is that neither model was trained on the MSCOCO caption that is the source of our evaluation benchmarks, thereby ensuring a fair comparison. Finally, **compositional reasoning REC**, which serves as our primary point of comparison, explicitly parses and executes the linguistic structure to localize referents through multi-step reasoning. Further details of experiments are provided in Appendix A.4.

## 4.2 MAIN RESULTS

We evaluate our framework along three key dimensions: (i) robustness in handling *no-target* cases (Section 4.2.1), (ii) efficiency in terms of failure rate and execution latency (Section 4.2.2), and (iii) scalability, highlighting the benefits of our decoupled pipeline (Section 4.2.3).

### 4.2.1 ROBUSTNESS ON NO-TARGET CASES

As shown in Table 2, proposal-based baselines yield near-zero TNR because they are effectively forced-prediction systems: from a pre-generated pool (e.g., Faster R-CNN in MAttNet (Ren et al., 2016; Yu et al., 2018)), they must return one region, and a candidate is almost always available. Detector-based and compositional methods further suffer from OVD hallucinations; even self-correction often forces a choice rather than enabling abstention, inflating FPR. This design reflects an implicit trade-off—optimizing TPR on standard benchmarks at the expense of no-target

Table 2: Comparison of REC methods on robustness using the gRefCOCO no-target testA split, and on standard REC performance using the RefCOCO testA split.

| Method | Balanced Acc. ↑ | No Target Robustness | | Standard REC |
| | | TNR (N-acc) ↑ | FPR ↓ | TPR (Acc@0.5) ↑ |
|---|---|---|---|---|
| *Fully Supervised REC (He et al., 2023)* | | | | |
| GREC-MDETR-R101 (Kamath et al., 2021) | 62.0 | 34.5 | 65.5 | 89.6 |
| GREC-UNINEXT-R50 (Yan et al., 2023) | 70.4 | 49.3 | 50.7 | 91.5 |
| *Proposal-based REC* | | | | |
| ReCLIP (Subramanian et al., 2022) | 23.1 | 0.0 | 100.0 | 46.1 |
| SS-CLIP (Han et al., 2024) | 33.3 | 0.0 | 100.0 | 66.5 |
| GroundVLP (Shen et al., 2024) | 30.7 | 0.0 | 100.0 | 61.3 |
| *Detector-based REC* | | | | |
| GroundingDINO-T (Liu et al., 2024) | 39.1 | 22.8 | 77.2 | 57.1 |
| GLIP-L (Li et al., 2022) | 37.2 | 21.7 | 78.3 | 52.6 |
| *Compositional Reasoning REC with GroundingDINO* | | | | |
| ViperGPT (Surís et al., 2023) | 33.4 | 0.2 | 99.9 | 66.7 |
| HYDRA (Ke et al., 2024) | 35.2 | 7.5 | 92.5 | 62.8 |
| NAVER (Cai et al., 2025) | 33.8 | 3.4 | 96.6 | 64.2 |
| VIRO (Ours) | **61.1** | **50.2** | **49.8** | **71.9** |

Table 3: Comparison of accuracy and efficiency for compositional reasoning methods on standard REC benchmarks. Accuracy is reported on the testA splits of RefCOCO/+, and the test split of Ref-COCOg, including excluding failure accuracy (Exc.↑) and include failure accuracy (Inc.↑). Failure rate (%) and FPS on execution stage are evaluated on RefCOCOg and RefCOCO, respectively.

| Method | Failure Rate ↓ | RefCOCO | | RefCOCO+ | | RefCOCOg | | FPS ↑ |
| | | Exc. | Inc. | Exc. | Inc. | Exc. | Inc. | |
|---|---|---|---|---|---|---|---|---|
| *Detector-based REC* | | | | | | | | |
| GroundingDINO-T (Liu et al., 2024) | 0.00 | 57.2 | 57.2 | 57.6 | 57.6 | 59.5 | 59.5 | 5.00 |
| GLIP-L (Li et al., 2022) | 0.00 | 52.6 | 52.6 | 48.6 | 48.6 | 52.6 | 52.6 | 1.23 |
| *Compositional Reasoning REC with GroundingDINO* | | | | | | | | |
| ViperGPT (Surís et al., 2023) | 6.03 | 66.7 | 64.4 | 61.7 | 57.5 | 65.7 | 61.7 | 0.67 |
| HYDRA (Ke et al., 2024) | 32.37 | 62.8 | 44.9 | 58.4 | 37.4 | 67.1 | 45.4 | 0.05 |
| NAVER (Cai et al., 2025) | 9.74 | 64.2 | 60.3 | 60.1 | 55.6 | **68.4** | 55.8 | 0.20 |
| VIRO (Ours) | **0.30** | **71.9** | **71.9** | **63.3** | **63.3** | 66.6 | **66.3** | **1.39** |
| *Compositional Reasoning REC with GLIP* | | | | | | | | |
| ViperGPT (Surís et al., 2023) | 6.31 | 72.0 | 69.6 | **65.7** | 61.3 | 69.6 | 65.2 | 0.68 |
| HYDRA (Ke et al., 2024) | 21.44 | 73.1 | 60.5 | 60.6 | 48.7 | 67.6 | 53.1 | 0.05 |
| NAVER (Cai et al., 2025) | 25.90 | **73.4** | 62.5 | 62.7 | 50.7 | **70.0** | 51.9 | 0.17 |
| VIRO (Ours) | **0.30** | 72.8 | **72.8** | 63.7 | **63.7** | 67.0 | **66.8** | 1.25 |

robustness—and can yield misleadingly high TPR by rewarding guesses. We illustrate this in Appendix A.5, where a forced-prediction variant of our method attains state-of-art TPR.

In contrast, VIRO attains 61.1% Balanced Accuracy substantially outperforming all baselines without REC fine-tuning. These gains stem from verification-integrated operators that enable abstention when no valid referent is present. Importantly, VIRO's robustness approaches that of fully supervised methods (e.g., GREC-UNINEXT) without training on no-target annotations, demonstrating robust zero-shot visual grounding.

### 4.2.2 ANALYSIS OF COMPOSITIONAL REASONING ON STANDARD REC BENCHMARKS

Table 3 provides a comprehensive evaluation of VIRO, against compositional reasoning REC baselines, focusing on its accuracy, execution efficiency, and throughput on standard REC benchmarks, using both GroundingDINO-T and GLIP-L. We report both Exc. (exclude failure accuracy) and Inc. (include failure accuracy), where the latter penalizes program failures as incorrect predictions, *i.e.*, evaluating all datasets.

VIRO shows a remarkably low program failure rate of 0.3%, which ensures that its Inc. accuracy remains nearly identical to its Exc. accuracy. In contrast, competing methods such as HYDRA and NAVER are prone to failures from two sources: (i) compile errors due to syntactically incorrect

programs generated by the LLM, and (ii) execution timeouts when an answer is not found within a default number of iterations (e.g., 7 for HYDRA, 5 for NAVER as default value). In contrast, VIRO's integrated validator, with a maximum of 5 iterations, ensures that nearly all generated programs are executable, overcoming a critical bottleneck of previous compositional approaches.

Furthermore, VIRO achieves superior computational efficiency. The results show that our method delivers high throughput (FPS) during the execution stage, significantly surpassing other compositional reasoning methods. This demonstrates that VIRO incorporates a sophisticated reasoning layer via its operators without imposing a heavy computational burden, making it highly suitable for processing a large number of images as analyzed in the following section.

### 4.2.3 SCALABILITY IN 1-QUERY–$N$-IMAGES

We evaluate the scalability of VIRO in a 1-query-$N$-images setting, where a single query is used to localize a target across multiple $N$ images. This scenario is crucial for real-world applications such as robotic visual search. As shown in Figure 3, VIRO and ViperGPT demonstrates exceptional scalability. Due its decoupled architecture, the program generation (pre-execution) is performed only once per query, *i.e.*, $T_{\text{total}} = T_{\text{pre-execution}} + N \times T_{\text{execution}}$. In contrast, methods like HYDRA and NAVER entangle these stages, forcing program regeneration for each image. Their total time is calculated as $T_{\text{total}} = N \times T_{\text{pre-execution}} + N \times T_{\text{execution}}$, leading to a non-linear growth in processing time that is impractical for large-scale tasks. This result highlights the critical advantage of our decoupled design for achieving low-latency visual reasoning at scale.

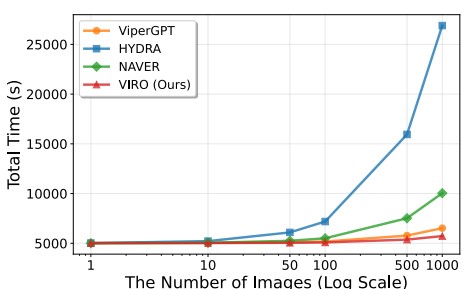

Figure 3: Total time in a 1-query-$N$-images setting, with the $x$-axis on a log scale.

### 4.3 ABLATION STUDIES

In this section, we conduct ablation studies to analyze the impact of key hyperparameters on VIRO's performance and to validate our design choices. Unless stated otherwise, all experiments are conducted using VIRO with GroundingDINO-T, evaluated on the RefCOCO testA split.

Table 4: Ablation study of the proposed verification components in VIRO. 'Fixed' refers to a fixed threshold, while 'adaptive' refers to an adaptive threshold.

| Method | Balanced Acc. ↑ | No Target Robustness | | Standard REC |
| | | TNR (N-acc) ↑ | FPR ↓ | TPR (Acc@0.5) ↑ |
| --- | --- | --- | --- | --- |
| Detector-only (GroundingDINO-T) | 39.1 | 22.8 | 77.2 | 57.1 |
| + Neuro-symbolic (w/o verification) | 56.8 | 38.9 | 61.1 | **74.6** |
| + Logical Verification (LV) | 57.0 | 39.3 | 60.7 | 74.6 |
| + Uncertainty Filtering (UF, fixed) | 58.5 | 43.1 | 56.9 | 74.4 |
| + Uncertainty Filtering (UF, adaptive) | **61.1** | **50.2** | **49.8** | 71.9 |

**Verification Components.** Table 4 shows the results of a cumulative ablation study to analyze the contribution of each verification module of our VIRO pipeline. We start with a standard detector-only baseline and progressively add our proposed modules to measure the impact on both no-target robustness and standard REC accuracy. By incorporating neuro-symbolic reasoning without verification, the performance improves significantly, with a balanced accuracy of 56.8%. This improvement stems from the compositional pipeline's ability to ground all noun phrases in the query, enabling sophisticated reasoning. The addition of LV and UF yields a further incremental improvement in no-target robustness, with a small reduction in TPR that reflects a precision–recall trade-off.

**OVD Detection Threshold.** The detection threshold of the Open-Vocabulary Detector (OVD) is a critical parameter that governs the trade-off between standard REC accuracy (TPR) and no-target robustness (TNR). As shown in Figure 4, a higher threshold improves TNR by filtering out spurious detections, but this simultaneously lowers recall, which in turn degrades TPR. We adopt a threshold of 0.2 to favor high recall at the proposal stage while maintaining balanced overall performance.

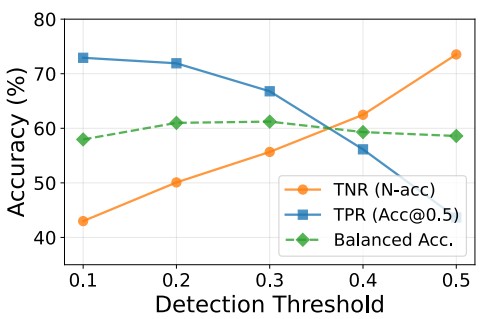

Table 5: Ablation study on early-exit evaluated on gRefCOCO no-target testA split.

| Early-exit | Latency ↓ | FPS ↑ |
|---|---|---|
| Enabled | 0.52 | 1.39 |
| Disabled | 0.58 | 1.79 |

Figure 4: Analysis of the OVD detection threshold, which illustrates the trade-off between TPR and TNR.

Table 6: Ablation study on the CLIP model.

| CLIP Model | TPR (Acc@0.5) ↑ | FPS ↑ |
|---|---|---|
| ViT-H/14 | 71.9 | 1.39 |
| ViT-L/14 | 68.8 | 1.79 |

**Description**: the dog on the street  **Description**: middle person

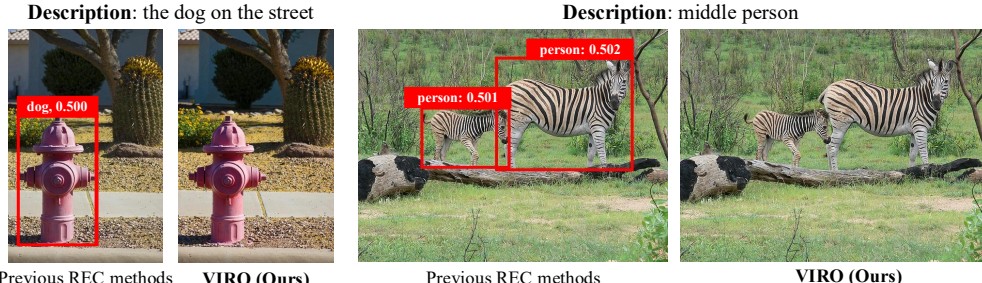

Previous REC methods   **VIRO (Ours)**   Previous REC methods   **VIRO (Ours)**

Figure 5: RefCOCO validation examples for false-positive suppression. Prior REC methods (left in each pair) produce spurious detections (red boxes), whereas VIRO (right) rejects them via CLIP-based verification, yielding no prediction in these cases.

**Early-exit Mechanism.**   As shown in Table 5, enabling early-exit reduces the average latency to 0.52 seconds per query on the gRefCOCO no-target testA split. Because operators run sequentially, unmet intermediate conditions trigger termination—for example, for "an elephant to the left of a man," the program exits if the elephant is not found, avoiding downstream work and boosting throughput when no-target cases are frequent.

**CLIP Models.**   We analyze the impact of the CLIP model backbone, used for both UF and the PROPERTY operator. Table 6 compares the performance of ViT-L/14 and ViT-H/14. Our final configuration use ViT-H/14 as it provides the best balance between accuracy and efficiency, although the lighter ViT-L/14 variant remains a computationally efficient alternative.

### 4.4 QUALITATIVE ANALYSIS

We present qualitative examples from RefCOCO validation set. Figure 5 demonstrates VIRO's ability to suppress false positives (FPs) from open-vocabulary detectors via CLIP-based verification, compared against previous REC methods. Illustrative examples of our program's execution process are provided in Appendix A.7.

### 5 DISCUSSION AND CONCLUSION

In this work, we introduced a verification-integrated neuro-symbolic pipeline VIRO for REC. By embedding lightweight verification mechanisms into each reasoning step, our framework explicitly handles no-target scenarios, achieves interpretable early termination, and maintains state-of-the-art accuracy with strong computational efficiency. Beyond REC, the modularity of our pipeline suggest promising extensions to interactive domains. In particular, the ability to parse natural language into verifiable symbolic programs opens the door to future applications where robots and humans engage in dialogue. In such scenarios, robots could not only interpret complex instructions but also execute them safely and transparently, ensuring that ambiguous or invalid commands are rejected before action. This direction underscores the broader potential of our approach as a foundation for trustworthy multimodal reasoning in embodied AI systems.

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
