# A  APPENDIX

We used a LLM as a general-purpose writing assistant for minor edits (clarity, grammar, or phrasing) and for generating alternative phrasings of paragraphs we had already drafted.

## A.1  DETAILS OF VIRO

### A.1.1  INPUT ARGUMENTS

We provide the details of VIRO by extracting the program generator prompts. The following functions are available in our framework for reasoning:

- **FIND(object_name='object_name')**: Returns all objects matching the object name which are clearly detectable, excluding non-object entities (e.g., living room, field, wall).
- **LOCATE(object=objects, position='location')**: Returns objects positioned at a specified absolute location, independent of other objects in the 2D space (e.g., 'right', 'at the bottom', 'on left', '9 o clock', 'outmost right', 'top', 'uppermost', 'middle', 'center').
- **ORDER(object=objects, criteria=['left'|'right'|'top'|'bottom'], rank=number)**: Returns the object positioned at the specified `rank` when sorted only by the given criteria ('left', 'right', 'top', 'bottom'), counting from the end.
- **ABSOLUTE_DEPTH(object=objects, criteria=['front'|'behind'])**: Returns objects from `objects` positioned absolutely closest (front) or farthest (behind) in the 3D space (depth information).
- **SIZE(object=objects, criteria=['big'|'small'])**: Returns objects filtered by relative size only by the given criteria ('big', 'small').
- **PROPERTY(object=objects, value='attribute')**: Filters objects based on their intrinsic attributes (e.g., color and patterns: 'red', 'striped', clothing: 'wearing a blue shirt', states or actions: 'standing', 'sitting', 'turned on', 'open').
- **FIND_DIRECTION(object=objects1, reference_object=objects2, criteria=['left'|'right'|'top'|'bottom'])**: Returns objects from `objects1` positioned next to objects in `objects2` only by the given criteria ('left', 'right', 'top', 'bottom').
- **FIND_NEAR(object=objects1, reference_object=objects2)**: Returns objects from `objects1` that are spatially close to any object in `objects2`.
- **FIND_INSIDE(object=objects1, reference_object=objects2)**: Returns objects from `objects1` that are strictly inside the reference object `objects2`.
- **RELATIVE_DEPTH(object=objects1, reference_object=objects2, criteria=['front'|'behind'])**: Returns objects from `objects1` positioned in depth relative to objects in `objects2` only by the given criteria ('front', 'behind').
- **RESULT(object=answer_object)**: Pre-processes the final selected object to the final answer form.

### A.1.2  VERIFICATION MODULE

**Relative Spatial Operators.** All relative spatial operators take at least two arguments: `object` and `reference_object`. As discussed in Section 3.2.1, we use logical verification in `RELATIVE_DEPTH`, similar to `FIND_DIRECTION`, but with the relative depth values 'front' or 'behind'. Additionally, in `FIND_INSIDE`, if there is no intersection area between the `object` and `reference_object`, the proposal is rejected.

**Attribute Operator.** We use the `PROPERTY` operator to filter visual attributes using the CLIP and GroundingDINO-T model. CLIP has varying thresholds depending on the given image, which makes filtering based solely on similarity scores challenging. To improve the filtering process, we first apply a softmax transformation to the CLIP similarity scores of the candidate regions from the OVD. We then integrate GroundingDINO-T scores into the filtering process, combining them with the softmax CLIP scores via a weighted sum. Finally, we set an adaptive threshold based on the number of candidates from `FIND` operator.

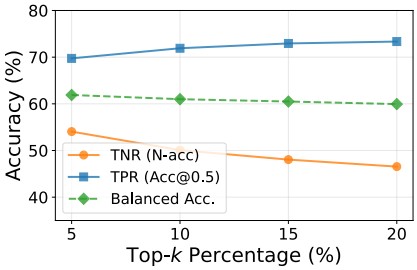

Figure A1: Analysis of $k$ adaptive threshold which illustrates the trade-off between TPR and TNR.

## A.2 ADAPTIVE THRESHOLD

To compute the filtering score $S(l|I_j)$, for each candidate region $I_j$ with label $l$, we leverage a pre-trained VLM trained with contrastive loss, such as CLIP. CLIP is designed to align visual and textual representations in a shared embedding space, enabling effective discrimination between semantically relevant (positive) and irrelevant (negative) pairs. However, one of the main challenges when using CLIP for verification is its inherent toward labels that appear frequently in its training data. Labels like 'person' or 'car' typically receive higher confidence scores compared to more specialized or rare objects, making a fixed threshold inappropriate for fair evaluation across different object categories. To address this label-specific bias, we implement an adaptive thresholding mechanism that calibrates the decision boundary for each target label individually. We utilize ImageNet as an auxiliary calibration dataset, computing verification scores for a representative sample of images containing various object categories.

We collect 5 images per class in ImageNet (a total of 5,000 images) and use Grounding DINO to crop out only the relevant class objects (denoted as $D_A$). This process helps minimize bias from background elements, where many images contain a person even when the target class is not "person." For each target label $l$, we analyze the distribution of verification scores $S(l|I)$ across this preprocessed dataset to determine CLIP's typical confidence range for that specific category.

The calibration process employs a top-$k$ selection strategy to determine the appropriate threshold $\delta_l$ Specifically, we rank all verification scores computed on the auxiliary dataset for target label $l$ where $d \in D_A$, and the score is defined as:

$$S(l\,|\,d_i) = \frac{1}{K}\sum_{k=1}^{K}\frac{\exp(\mathrm{sim}(d_i,l)/\tau)}{\exp(\mathrm{sim}(d_i,l)/\tau)+\exp(\mathrm{sim}(d_i,c_k)/\tau)}\;. \tag{4}$$

The set of verification scores for the entire dataset is represented as:

$$\mathcal{S}(l|D_A) = \{S(l|d_1), S(l|d_2), \dots, S(l|d_n)\}\;, \tag{5}$$

where $n$ is the number of auxiliary dataset. The threshold $\delta_l$ select the top-$k$% highest scoring samples, where $k$ is a hyperparameter (typically set as 10). This strategy effectively captures the performance level that CLIP consistently achieves for high-confidence predictions of the target category, providing a data-driven threshold that reflects CLIP's inherent capability for that specific label.

We also investigate the sensitivity $k$ of our pipeline. As shown in Figure A1, VIRO's performance remains highly robust to changes in the Top-k percentage. Although a slight increase in TPR is observed with more candidates (as the ground-truth box is more likely to be included), the overall Balanced Accuracy stays stable across different values of $k$.

## A.3 PROGRAM VALIDATOR

The Program Validator serves as a safeguard against errors in LLM-generated programs. Its primary purpose is to detect syntactic, structural, and logical mistakes before program execution, and to provide structured feedback tailored to the specific type of error. Since LLM outputs are not guaranteed to be flawless, the validator detects invalid programs at pre-execution time and provides feedback that guides the LLM to regenerate a correct program.

A valid program must follow a simple and strict format: each line is written as `VAR = OP(ARG=..., ...)` and executed sequentially in order, while the program always terminates with a line that must take the form `FINAL_RESULT = RESULT(object=VAR)`. Given this constrained design, the validator enforces the following properties:

- **Syntax enforcement:** all lines except the last must take the form `VAR = OP(ARG=..., ...)`, and the last line must take the form `FINAL_RESULT = RESULT(object=VAR)`.

- **Variable tracking:** at line $t$, only variables defined in lines $1, \ldots, t-1$ may be referenced, and redefinition of an existing variable is disallowed to prevent overwriting outputs from earlier steps.

- **Argument typing:** arguments are checked for valid type (variable, string, or number). Certain arguments have additional constraints, e.g., `rank` must be a positive integer and `criteria` must be chosen from a predefined set (e.g., `left`, `right`, `top`, `bottom`).

- **Operator constraints:** each operator must belong to the predefined functions (e.g., `FIND`, `FILTER`), and each function must include all of its required arguments.

- **Output format:** the program must end with a single line in the form `FINAL_RESULT = RESULT(object=VAR)`, and no other `RESULT` may appear earlier in the program.

Through these checks, the Program Validator acts as a reliable filter and feedback mechanism, reducing execution errors caused by incorrect code generated by the LLM. Both our method and ViperGPT (Surís et al. (2023)) use programs generated by the LLM prior to execution, but the scope of validation differs fundamentally. ViperGPT only ensures that the generated code is syntactically valid Python code, which allows issues such as undefined variables, incorrect argument types, or missing required arguments to pass through unnoticed until runtime. In contrast, our validator applies strict structural and semantic checks before execution, ensuring correct format, valid variables, and a proper output statement. These checks reduce execution errors and provide targeted feedback for correction.

## A.4 DETAILS OF EXPERIMENTS

### A.4.1 DATASETS

**gRefCOCO no-target split.** gRefCOCO is a referring expression dataset that explicitly includes no target queries, allowing evaluation of systems that must either localize the referred region or abstain when no valid target exists (He et al., 2023; Liu et al., 2023). To avoid trivial negatives, no-target expressions are constrained to be contextually related to the image, and annotators may reuse deceptive expressions from the same split when needed.

**RefCOCO/+/g.** RefCOCO (Yu et al., 2016) primarily targets location-based expressions, while RefCOCO+ (Yu et al., 2016) focuses on attribute-based descriptions by prohibiting the use of absolute location words. RefCOCOg (Mao et al., 2016), in contrast, contains longer and more complex expressions, often combining both spatial relations and attributes. For RefCOCO and RefCOCO+, results are reported separately on two test splits: TestA, which includes images with people as referents, and TestB, which includes images with objects other than people. All of these benchmarks are built upon the MSCOCO (Lin et al., 2014) image dataset.

### A.4.2 EVALUATION METRICS

We assess both *no-target robustness* and standard REC accuracy, which jointly require addressing localization and classification. Following a binary classification view, we define outcomes based on the confusion matrix: **True Positive (TP)**: a target is present and the model correctly localizes it (Intersection-over-Union (IoU) $> 0.5$); **True Negative (TN)**: target is absent and the model correctly predicts its absence; **False Positive (FP)**: target is absent but the model incorrectly outputs a bounding box; and **False Negative (FN)**: a target is present but the model either predicts 'no target' or localizes it incorrectly (IoU $< 0.5$).

### A.4.3 BASELINE DETAILS

**Proposal-based REC** first parse the referring expression to isolate key linguistic components before matching them against pre-generated object proposals. **ReCLIP** (Subramanian et al., 2022) employs a syntactic parser to extract noun chunks, while **GroundVLP** (Shen et al., 2024) utilizes a traditional NLP toolbox to identify the main object. More recent approaches leverage the advanced capabilities of Large Language Models (LLMs) for this task; **SS-CLIP/SS-FLAVA** (Han et al., 2024) (Chen & Chen, 2025) uses an LLM to parse the main object from the query. After this initial parsing stage, each method employs its unique mechanism to map the extracted components to the most relevant regions in given detected proposals from MAttNet (Yu et al., 2018). These are architecturally constrained to a rich pool of candidate regions by Faster RCNN, making a direct comparison on target-absent task unfair.

**Compositional reasoning REC** parse complex queries into explicit programs. By generating and then executing these programs, they transparently handle multi-step compositional logic to derive the final result. We compare our approach with state-of-the-art methods in this domain, including **ViperGPT** (Surís et al., 2023), **HYDRA** (Subramanian et al., 2022), and **NAVER** (Cai et al., 2025), to benchmark its compositional reasoning capabilities.

### A.4.4 IMPLEMENTATION DETAILS.

We primarily follow the official implementations of each baseline, using their default hyperparameter settings. Unless otherwise noted, detection thresholds for open-vocabulary detectors are fixed based on validation performance on RefCOCO: $0.2$ for GroundingDINO-T and $0.5$ for GLIP-L. For all program generation, we use Qwen2.5-72B-Instruct-AWQ (Team, 2024), known for strong code-generation, ensuring a fair comparison with Python-code baselines such as ViperGPT.

### A.5 STANDARD REC ACCURACY UNDER FORCED PREDICTION

Table A1: Accuracy comparison on the referring expression detection task RefCOCO/+/g datasets.

| Method | RefCOCO | | | RefCOCO+ | | | RefCOCOg | |
|---|---|---|---|---|---|---|---|---|
| | Val | TestA | TestB | Val | TestA | TestB | Val | Test |
| MDETR | 86.75 | 89.58 | 81.41 | 79.52 | 84.09 | 70.62 | 81.64 | 80.89 |
| UNINEXT | 89.72 | 91.52 | 86.93 | 79.76 | 85.23 | 72.78 | 83.95 | 84.31 |
| ReCLIP | 45.8 | 47.0 | 45.2 | 45.3 | 48.5 | 42.7 | 57.0 | 56.2 |
| SS-CLIP | 60.6 | 66.5 | 54.9 | 55.5 | 62.6 | 45.7 | 59.9 | 59.9 |
| SS-FLAVA | 52.5 | 52.7 | 52.9 | 50.8 | 53.4 | 47.6 | 61.3 | 60.9 |
| GroundVLP | 52.6 | 61.3 | 43.5 | 56.4 | 64.8 | 47.4 | 64.3 | 63.5 |
| GroundingDINO-T | 50.4 | 57.2 | 43.2 | 51.4 | 57.6 | 45.8 | 60.4 | 59.5 |
| ViperGPT | 62.2 | 66.7 | 54.6 | 55.4 | 61.7 | 50.4 | 66.0 | 65.7 |
| NAVER | 61.1 | 64.2 | 58.2 | 56.4 | 60.1 | 51.8 | 68.4 | 68.4 |
| GDINO-FLORA$^\dagger$ | 73.7 | 78.5 | 67.8 | 63.2 | 71.6 | 53.5 | 72.5 | 72.1 |
| VIRO (Ours) | 71.4 | 75.0 | 64.4 | 59.5 | 65.8 | 50.1 | 69.6 | 70.3 |
| GLIP-L | 47.5 | 52.6 | 41.8 | 44.1 | 48.6 | 39.8 | 51.9 | 52.6 |
| ViperGPT | 66.9 | 72.0 | 59.9 | 59.6 | 65.7 | 63.0 | 69.3 | 69.6 |
| HYDRA | 68.0 | 73.1 | 62.5 | 55.8 | 60.6 | 50.6 | 67.2 | 67.6 |
| NAVER | 69.6 | 73.4 | 64.4 | 59.0 | 62.7 | 56.4 | 70.7 | 70.0 |
| VIRO (Ours) | 71.4 | 75.7 | 63.8 | 59.3 | 66.2 | 51.3 | 70.6 | 71.5 |

$^\dagger$ Official code has not been released as of September 25, 2025.

### A.6 ANALYSIS OF SELF-CORRECTION OF NAVER

Table A2 compares the performance of the NAVER framework with and without its self-correction mechanism against VIRO. The Perceptioner, Logic Reasoner, and Logic Answerer modules in NAVER trigger self-correction when no valid target is found, forcing an object prediction. We evaluate the performance on the no-target dataset by disabling forced self-correction in the three

modules and instead applying Early-Exit. We measure TNR and FPR on the gRefCOCO testA no-target dataset, and TPR on the RefCOCO testA dataset.

*Early-Exit Strategies.* Perceptioner Early-Exit (P Early-Exit): Stops inference if the open-vocabulary detector finds no objects, preventing self-correction. Logic Reasoner Early-Exit (LR Early-Exit): Stops inference if geometric relations among candidate objects are logically invalid. Logic Answerer Early-Exit (LA Early-Exit): Stops inference if a high-capacity MLLM rejects the final predicted object as inconsistent with the query.

Table A2: Performance comparison of the NAVER framework (w/ and w/o self-correction) and VIRO (Ours).

| Method | No Target Robustness | | | Standard REC |
|---|---|---|---|---|
| | Balanced Acc ↑ | TNR ↑ | FPR ↓ | TPR ↑ |
| NAVER w/ self-correction | 33.8 | 3.4 | 96.6 | 64.2 |
| NAVER w/o self-correction | 63.2 | 71.6 | 28.4 | 54.8 |
| VIRO (Ours) | 61.1 | 50.2 | 49.8 | 71.9 |

Table A3: Frequency of Early-Exit Types of NAVER and Overall Result.

| Frequency of Early-Exit Types of NAVER | | | Overall Result | | |
|---|---|---|---|---|---|
| P Early-Exit | LR Early-Exit | LA Early-Exit | TN | FP | Total |
| 1998 (67.8%) | 344 (11.7%) | 604 (20.5%) | 2946 (71.6%) | 1168 (28.4%) | 4114 |

While VIRO shows slightly lower Balanced Accuracy, it avoids NAVER's heavy LA Early-Exit module, which causes significantly longer execution time (see Table **??**).

## A.7 EXECUTION EXAMPLES OF VIRO

**Query**: the blade thing on the top jet

```
OBJ0 = FIND(object_name="blade")
OBJ1 = FIND(object_name="jet")
ANSWER0 = FIND_DIRECTION(object=OBJ0, reference_object=OBJ1, criteria="top")
FINAL_RESULT = RESULT(object=ANSWER0)
```

Figure A2: Program generated for the query "the blade thing on the top jet" (top), along with its sequential execution from step 1 to step 4 (bottom).

**Query**: cup to the top right of muffin

```
OBJ0 = FIND(object_name="cup")
OBJ1 = FIND(object_name="muffin")
OBJ2 = FIND_DIRECTION(object=OBJ0, reference_object=OBJ1, criteria="top")
ANSWER0 = FIND_DIRECTION(object=OBJ2, reference_object=OBJ1, criteria="right")
FINAL_RESULT = RESULT(object=ANSWER0)
```

Figure A3: Program generated for the query "cup to the top right of muffin" (top), along with its sequential execution from step 1 to step 4 (bottom).

**Query**: man at head of table white shirt closest

```
OBJ0 = FIND(object_name="man")
OBJ1 = FILTER(object=OBJ0, value="wearing white shirt")
OBJ2 = FIND(object_name="table")
OBJ3 = FIND_DIRECTION(object=OBJ1, reference_object=OBJ2, criteria="top")
ANSWER0 = ABSOLUTE_DEPTH(object=OBJ3, criteria="front")
FINAL_RESULT = RESULT(object=ANSWER0)
```

Figure A4: Program generated for the query "man at head of table white shirt closest" (top), along with its sequential execution from step 1 to step 6 (bottom).