# OpenReview forum: "VIRO: Efficient and Robust Neuro-Symbolic Reasoning with Verification for Referring Expression Comprehension"
_ICLR.cc/2026/Conference — ICLR 2026 Conference Withdrawn Submission_

### Official Review · Reviewer_RR4e · 2025-10-20

**Soundness:** 3
**Presentation:** 3
**Contribution:** 3
**Rating:** 6
**Confidence:** 4

**Summary:**

- The authors propose VIRO — Verification-Integrated Reasoning Operators, a neuro-symbolic framework for Referring Expression Comprehension (REC) to solve the task of identifying an image region that corresponds to a natural-language description.

- VIRO bridges symbolic logic and neural vision-language reasoning to create an efficient, interpretable, and verifiable REC system.

- VIRO improves robustness (especially in “no target” scenes) and computational efficiency, key for real-time applications like robot perception and multimodal AI systems.

**Strengths:**

### 1. Technical Strengths

- Each reasoning operator (e.g., FIND, FIND_DIRECTION) has built-in verification, enabling self-checking and early termination — a novel design in neuro-symbolic reasoning.

- Unlike prior models that always predict a box, VIRO can abstain when no valid referent exists, increasing robustness and realism.

- CLIP-based Uncertainty Filter: Efficiently removes false positives from open-vocabulary detectors with minimal overhead.

- Logical spatial verification ensures relational consistency (“left of”, “inside”, etc.) through simple geometric checks.

- Decoupled pipeline separates program generation (by LLM) from execution, allowing one query to be reused across multiple images — improving latency and scalability.

- Neuro-Symbolic integration combines the interpretability and modularity of symbolic reasoning with the perception strength of neural models (OVD + CLIP + DepthAnything).

- Program validation & grammar checking ensures syntactic correctness of generated programs before execution, avoiding runtime errors common in prior systems like HYDRA and NAVER.

- Early-Exit mechanism stops reasoning when conditions fail (e.g., missing object), saving compute and improving responsiveness.

- The modular design could extend to other tasks — e.g., visual question answering, grounded instruction following, or embodied agents.

### 2. Experimental Strengths

- The authors have benchmarked their proposed methods on both standard (RefCOCO/+/g) and no-target (gRefCOCO) datasets. Evaluates robustness, efficiency, and scalability. The reported results are strong. For example,  61.1% balanced accuracy on no-target cases (state-of-the-art for zero-shot). 71.9% standard REC accuracy, outperforming compositional baselines. 0.3% failure rate vs. 6–32% for competing neuro-symbolic methods. Fast execution (1.39 FPS) and reduced latency through early exits.

- Clear component-level validation showing contributions of verification modules, CLIP thresholds, and early-exit design.

- Quantifies trade-offs between TPR (precision) and TNR (robustness).

- Scalability test (1-query–N-images) demonstrates linear scalability due to decoupled design; major advantage for multi-image tasks like robot search.

- Visual examples clearly show suppression of false positives — interpretable verification behavior.

**Weaknesses:**

### 1. Technical Limitations

- Performance depends on external pretrained models (GroundingDINO, CLIP, DepthAnything). Errors in these propagate to reasoning steps.

- Operators handle basic spatial and attribute reasoning, but not more complex temporal or abstract relations (e.g., “the man looking at the dog”).

- CLIP-based uncertainty filter uses threshold calibration (even adaptive per-label), which may not generalize across domains or unseen categories.

- Requires a large language model for program generation. Although decoupled, quality still depends on LLM prompt engineering and examples.

- Fixed operator grammar may limit reasoning flexibility compared to free-form program generation (e.g., ViperGPT’s Python-like reasoning).

- CLIP’s training biases (ImageNet-like categories) may skew verification toward frequent object types, missing rare or domain-specific terms.

### 2. Experimental Limitations

- Lack of Cross-Domain Evaluation: Tests limited to COCO-derived datasets; robustness on unseen domains (e.g., indoor scenes, robotics) not demonstrated.

- Limited Real-World Deployment Evidence: Efficiency gains shown in simulation (FPS, latency), but no evidence of integration into real-time or robotic systems.

- No Human Evaluation: While interpretability is claimed, no human studies assess clarity or usefulness of the generated reasoning traces.

- Balanced Accuracy Metric Only: Uses balanced accuracy as the key metric; other useful measures (precision-recall, F1, calibration error) not explored.

- Limited Comparison Scope: Although major baselines (HYDRA, NAVER, ViperGPT) are included, newer multimodal transformers (e.g., GPT-4V, Gemini) not tested.

**Questions:**

See the weakness section. In addition:

- How easily can the VIRO framework incorporate new reasoning operators (e.g., temporal, causal, or commonsense relations)?

- Would adding such operators require retraining or re-prompting the LLM?

- The CLIP-based uncertainty filter uses adaptive thresholds — how sensitive is performance to these threshold settings?

- Could self-calibrating or Bayesian uncertainty estimation methods improve reliability?

- How does VIRO handle ambiguous or under-specified queries (e.g., “the man near the big thing”)?

- Does the LLM's program generation incorporate any self-check or consistency verification?

- Why was CLIP chosen for verification instead of more specialized models like SigLIP or ImageBind?

- Have you considered multi-modal confidence fusion across detectors instead of relying on CLIP similarity scores alone?

- Given the reported FPS (~1.4), how feasible is VIRO for real-time applications like robotic perception or human-robot dialogue?

- What are the computational bottlenecks (e.g., detector inference, CLIP verification, or symbolic execution)?

- The paper emphasizes decoupling of program generation from execution. Are there cases where query-specific context (e.g., image content) should influence program generation dynamically?

- Could fully static programs miss context-dependent reasoning patterns?

- How does VIRO scale when handling long, compositional, or nested referring expressions (e.g., “the second man from the left holding a red cup behind the woman”)?

- Are there limits on program depth or recursion?

- Since the Open-Vocabulary Detector (OVD) may hallucinate unseen objects, could VIRO’s verification be misled by dataset bias or object frequency imbalance?

- Have you observed systematic failures across specific object types (e.g., small, occluded, or rare objects)?

- How much does the LLM’s choice (e.g., GPT-4 vs. Qwen2.5 vs. LLaMA) affect program structure, syntax errors, or reasoning efficiency?

- Balanced Accuracy is used as the main metric — did you also assess calibration, abstention accuracy, or false discovery rate for no-target handling?

- How does VIRO’s abstention threshold affect trade-offs between precision and recall?

---

### Official Review · Reviewer_mXRX · 2025-11-01

**Soundness:** 2
**Presentation:** 2
**Contribution:** 1
**Rating:** 4
**Confidence:** 3

**Summary:**

This paper proposes to add an extra filtering step to the output object list of referring expression compression (REC), to make sure no mismatches are returned. It claims to outperform existing REC methods with a large margin on the gRefCoCo no-target dataset.

**Strengths:**

1. The presentation is clear and easy to follow.

**Weaknesses:**

1. The added extra filtering step on the (image crop, caption) pair by CLIP, is kind of trivial. Moreover, the proposed method is based on a manually written program, which is a step backward from the trending end-to-end approaches.
2. The reported performance of baselines is very low on gRefCOCO no-target. For example, NAVER has an N-acc (no-target accuracy) of 3.4%. I find this result dubious, since in the code of NAVER, the authors explicitly checked the results returned by the logic reasoner. If there is no match, it will return an empty list:
https://github.com/ControlNet/NAVER/blob/4b0ae074701e57882a8b712bac09e15a73f005bc/naver/agent/logic_reasoning/logic_reasoner.py#L59 \
I don't see why it would achieve such a low N-acc.

**Questions:**

N/A

---

### Official Review · Reviewer_8mAU · 2025-11-01

**Soundness:** 2
**Presentation:** 2
**Contribution:** 2
**Rating:** 4
**Confidence:** 3

**Summary:**

This paper proposes VIRO, which is a neuro-symbolic method for referring expression comprehension that can clearly tell when the target object does not exist. It turns the input sentence into a short program made of step-by-step checks. If any check fails, the system stops early and outputs “no target” instead of forcing a wrong bounding box.

**Strengths:**

The designed each operator verifies its own result and can abort the pipeline when evidence is insufficient.
The method requires no task-specific fine-tuning or new datasets, lowering engineering overhead and easing deployment.

**Weaknesses:**

This paper is of limited novelty and motivation. Program-based visual question answering have been studied in prior program-execution works[1,2]. The paper positions verification as a first-class element but does not clearly establish a more fundamental new method.

The methodological contribution is limited. The approach mainly adds operator-level verification and early-exit checks to an not novel program pipeline. There is no new learning/training objective, or architecture novelty but more as engineering refinements.

The overall system relies on open-vocabulary detectors and other fundational models, so their errors and biases propagate through the pipeline which is severe for OOD cases(for fundamental models) as the system is training-free. Also the evaluation is centered on RefCOCO and gRefCOCO.

[1] ViperGPT: Visual Inference via Python Execution for Reasoning [2] Visual Programming: Compositional visual reasoning without training

**Questions:**

What specific capability does VIRO have more than previous methods which could not achieve?
Could the authors please provide evaluations or evidence on more diverse benchmarks to validate generalization?

---

### Official Review · Reviewer_BrL1 · 2025-11-02

**Soundness:** 2
**Presentation:** 3
**Contribution:** 2
**Rating:** 4
**Confidence:** 3

**Summary:**

The paper presents a neurosymbolic approach that addresses the problem of referring expression comprehension (REC) with compositional prompts and to enable efficient processing of cases wherein a referring target is not present in the image (which conventional approaches still allocate compute towards trying to solve and have high-confidence false positive rates). The authors use a few-shot prompted LLM to process an input REC-phrase into a symbolic program (for which operators are defined by authors), then a lightweight CLIP-based uncertainty filtering to identify cases wherein a REC phrase object is not present, and existing task-specific models for each defined symbolic operation. Experiments on REC benchmarks show the proposed method reduces false positive rates and error rates compared to existing compositional methods and baselines.

**Strengths:**

1. The approach is useful to reduce false positive rates and unnecessary compute on REC cases wherein the phrase is not present in the image.
2. Experiments show beneficial results of the method in reducing false positive rates and improving true positive rates compared to existing program-based compositional reasoning baselines. The method is also more robust and has a lesser failure rate than existing compositional reasoning baselines.
3. Ablations and additional analysis on early stopping, qualitative cases and detection thresholds provide useful results regarding the method.

**Weaknesses:**

1. The method appears to be less scalable and restricted to only REC than existing compositional reasoning methods such as ViperGPT and HYDRA which can work beyond REC tasks and for general vocabulary. Specifically, the method's dependence on predefined bank of K common categories (L210) represented in CLIP, make it unclear how it can be utilized for out-of-vocab settings. Would it not exhibit failure rates in these cases compared to open-domain methods such as ViperGPT?

2. The method's primary novelty appears to be in usage of CLIP-based uncertainty filtering and designing a set of symbolic operators specific to REC. While this does address the high false-positive rates, I don't see why the same cannot be done for ViperGPT as an additional step and to have additional checks exclusive to REC in their python programs API. The novelty in this regard appears low and also constrained to a specific setting in REC.

**Questions:**

Please see weaknesses above.
Additionally:
1) Is the LLM used the same as in ViperGPT for fair comparison? Ideally same LLM should be used for fair comparison so that performance improvements and error rates are not due to underlying LLM.

---

### Author Response · Authors · 2025-11-14

We have decided to withdraw this submission from consideration. We sincerely appreciate the reviewers’ constructive feedback and their efforts in evaluating our work. Their comments have provided valuable insights that will greatly help us strengthen this work in future revisions.

---

### Note · Authors · 2025-11-14

**Comment:**

We have decided to withdraw this submission from consideration. We sincerely appreciate the reviewers’ constructive feedback and their efforts in evaluating our work. Their comments have provided valuable insights that will greatly help us strengthen this work in future revisions.

**Withdrawal Confirmation:**

I have read and agree with the venue's withdrawal policy on behalf of myself and my co-authors.